# A New Wall Current Transformer for Accurate Beam Intensity Measurements in the Large Hadron Collider

Michal Krupa *[iD] and Marek Gasior *

Beam Instrumentation Group, Accelerator Systems Department, the European Organization for Nuclear Research (CERN), 1211 Geneva, Switzerland
* Correspondence: michal.krupa@cern.ch (M.K.); marek.gasior@cern.ch (M.G.)

**Abstract:** The Large Hadron Collider (LHC) stores two high-energy counter-rotating particle beams consisting of multiple bunches of a nanosecond length. Precise knowledge of the number of particles within each bunch, known as the bunch intensity, is crucial for physicists and accelerator operators. From the very beginning of the LHC operation, bunch intensity was measured by four commercial fast beam current transformers (FBCTs) coupling to the beam current. However, the FBCTs exhibited several shortcomings which degraded the measurement accuracy below the required level. A new sensor, the wall current transformer (WCT), has been developed to overcome the FBCT limitations. The WCT consists of eight small radio frequency (RF) current transformers distributed radially around the accelerator's vacuum chamber. Each transformer couples to a fraction of the image current induced on the vacuum chamber by the passing particle beam. A network of RF combiners sums the outputs of all transformers to produce a single signal which, after integration, is proportional to the bunch intensity. In laboratory tests and during beam measurements, the WCT performance was demonstrated to convincingly exceed that of the FBCT. All originally installed FBCTs were replaced by four WCTs, which have been serving as the LHC reference bunch intensity sensors since 2016.

**Keywords:** beam instrumentation; beam intensity measurements; current transformers

## 1. Introduction

The Large Hadron Collider (LHC) at CERN is the largest and highest-energy particle accelerator in the world [1]. It has a circumference of twenty seven kilometres and accelerates two counter-rotating particle beams to energies as high as 6.8 TeV. The beams travel for the most part in two separate vacuum chambers which are joint only close to the four large physics experiments where the beams cross each other, the collisions take place, and the paths and momenta of newly created particles are determined.

The LHC accelerates the beams of protons or positively charged ions of heavier elements. The beams are ultra-relativistic (the Lorentz factor exceeding 7200 for proton beams) and they travel very close to the speed of light (over 99.999999% *c* for proton beams).

As the particles flow around the accelerator, this motion of electric charge constitutes an electric current referred to as the beam current. Consequently, the sum of electric charges contained within a beam is called the beam charge. Another closely related beam parameter is its intensity, which is defined as the number of particles contained within the beam and can be simply calculated by dividing the beam charge by the elementary charge.

The beam particles do not form a continuous stream but are longitudinally grouped into discrete packets called bunches. The distribution of particles within a bunch is approximately Gaussian with a standard deviation ranging from 200 to 400 ps. A quantity which is typically used to describe the longitudinal bunch shape is the bunch length, assumed to be four standard deviations of the distribution.

The LHC revolution period of 88.925 µs is divided into 3564 bunch slots, each of which is approximately 25 ns long. However, the beam production mechanism and equipment

safety aspects require that approximately 20% of the available bunch slots are free of particles. The LHC beams consist of up to some 2800 bunches which are interleaved with empty bunch slots following complicated filling patterns. One complete revolution of the beam around the LHC is often referred to as a "turn".

Each bunch is characterised by its own current, charge and intensity. In most cases, the LHC uses two kinds of bunches: pilot bunches with an intensity of $5–10 \times 10^9$ ppb (protons per bunch) and nominal bunches with an intensity of $1–2.2 \times 10^{11}$ ppb. The corresponding peak bunch currents are approximately 1–2 A and 20–48 A, respectively. Figure 1 illustrates a hypothetical LHC bunch pattern and the turn numbering convention. Continuous and accurate measurements of the intensity of each bunch on a turn-by-turn basis are required to optimise the operation of the LHC (e.g., intensities of individual bunches should be similar), to ensure LHC equipment safety (e.g., maximal bunch intensity must be controlled), and to properly analyse the outcomes of high-energy collisions (e.g., individual intensities of the colliding and not colliding bunches must be known).

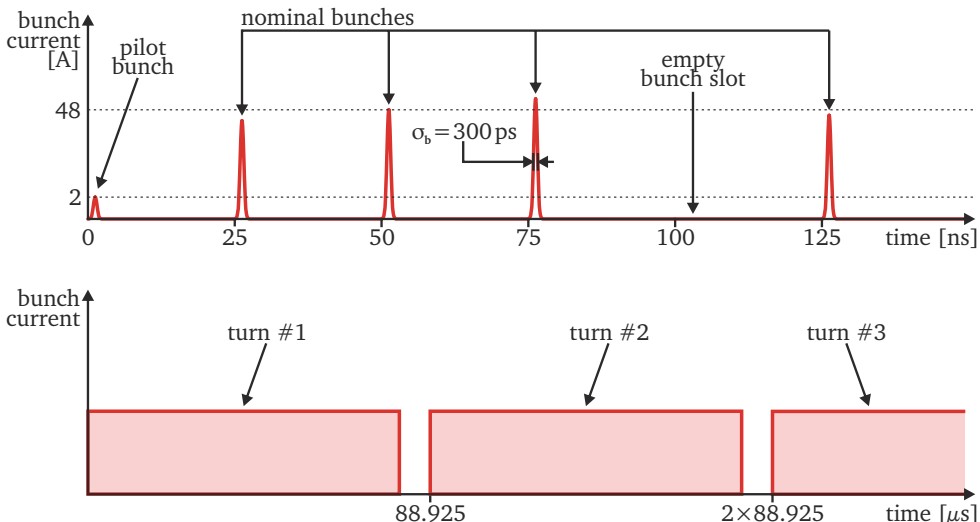

**Figure 1.** Temporal structure of the LHC beam.

Precise measurements of beam and bunch intensities are particularly crucial for calculating the luminosity $\mathcal{L}$ at the beam–beam collision points. The knowledge of this fundamental parameter is necessary to determine whether the outcomes of high-energy collisions are within the limits defined by the existing models or a new physics phenomenon has been discovered. For colliders, such as the LHC, the luminosity is given by:

$$\mathcal{L} = \frac{N_1 N_2 f_{\text{rev}}}{4\pi \sigma_x \sigma_y} \tag{1}$$

where $N_1$ and $N_2$ are the intensities of the two respective beams (or bunches), $f_{\text{rev}}$ is the frequency at which the beams (or bunches) collide, and $\sigma_x$ and $\sigma_y$ are the width and height of the effective overlap region of the two beams (or bunches) in the transverse plane. The beam and bunch intensity measurement error directly propagates to the luminosity calculation error, and therefore, the performance of a collider is strictly linked to the quality of its beam intensity monitoring system.

Beam and bunch intensity can be measured using one of the many techniques developed and applied in accelerators over the years [2]. In the LHC, both those quantities are measured with current transformers which couple to the electromagnetic field carried by the beam. In the first years of the LHC operation, the bunch intensity was monitored with the fast beam current transformers (FBCT) [3], which are the most widely used sensor for this purpose. A total of four FBCTs were in service, two redundant sensors for each LHC beam. However, the FBCTs exhibited an undesired sensitivity to the transverse beam position and an excessively long

output pulse which prevented measurements with sufficient accuracy [4]. Both these shortcomings limited the LHC FBCT performance to such an extent that studies were launched to find an alternative solution. This manuscript describes a new sensor resulting from these studies carried out within the framework of a Ph.D. thesis [5], the wall current transformer (WCT), optimised for accurate measurements of the bunch intensity in the LHC. This manuscript summarises the work and shows the most important measurements to disseminate this emerging technology within a wider community. The presented results are supported by seven years of successful and reliable operation in the LHC. The developed technology can find applications in other accelerators and be further optimised.

## 2. LHC Bunch Intensity Measurements with the FBCT

The instantaneous bunch current $i_B(t)$ and the bunch charge $Q_B$ are closely related to the bunch intensity $N_B$ and can be used for its indirect measurement. Integrating $i_B(t)$ over the duration of a bunch slot $T_B$ results in $Q_B$ which, after dividing by the elementary charge $e_0$, gives $N_B$:

$$N_B = \frac{1}{e_0} Q_B = \frac{1}{e_0} \int_{t_0}^{t_0 + T_B} i_B(t)\, dt \tag{2}$$

Therefore, the LHC bunch intensity can be calculated by continuously measuring the instantaneous bunch current and integrating it over windows of approximately 25 ns, corresponding to the distance between two consecutive bunches.

Moreover, it is not needed to measure the full frequency spectrum of the bunch current. A low-pass filter does not alter the signal's integral if its insertion loss at DC is negligible [5]. A bunch current pulse stretched by a low-pass filter can still serve as a basis for bunch intensity measurements as long as the pulse does not extend beyond the 25 ns window.

From the very first days of the LHC operation, the bunch intensity was measured with four commercial fast beam current transformers (FBCTs) [3], two per LHC ring. Similar devices are widely used in other particle accelerators but are more generally referred to as AC current transformers (ACCTs). Figure 2 illustrates their principle of operation.

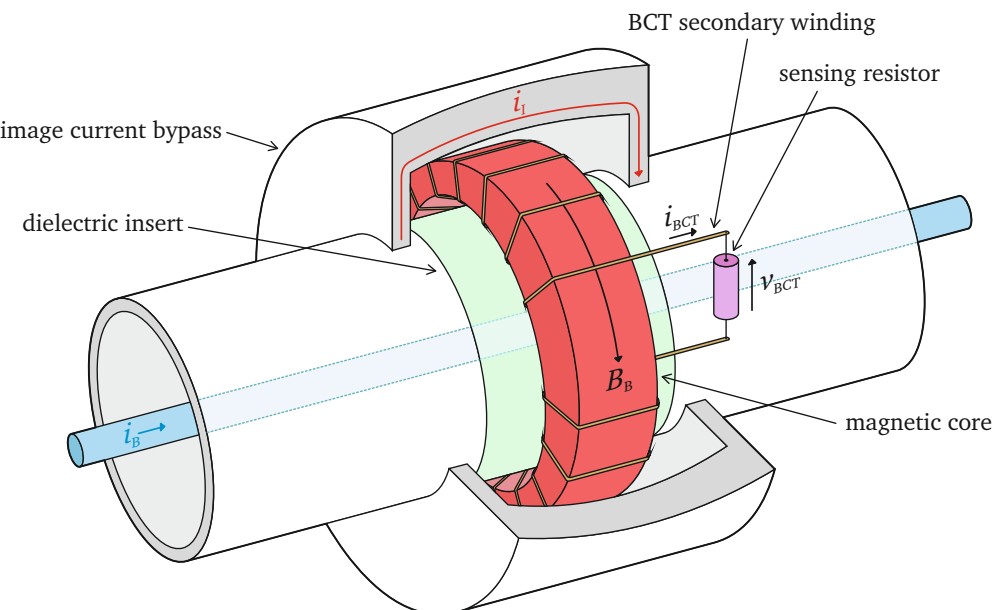

**Figure 2.** Principle of operation of the fast beam current transformer (FBCT) [5].

The FBCT employs a high-permeability toroidal magnetic core, which is installed over a dielectric insert. The conducting walls of the accelerator's vacuum chamber enclose the electromagnetic field carried by the charged beam in motion. The dielectric insert serves as a "window" through which the core can couple to the beam's field. Due to electromagnetic induction, this coupling results in a current flow through the secondary

wire wound around the core. For ultra-relativistic beams travelling close to the speed of light, such as the LHC beams, the induced current $i_{BCT}$ is proportional to the passing beam current $i_B$. This is a consequence of the relativistic Lorentz contraction phenomenon, illustrated in Figure 3, which shortens the longitudinal component of the field lines as the field's source approaches the speed of light. Consequently, the secondary current $i_{BCT}$ has the same temporal structure as the instantaneous beam current $i_B$.

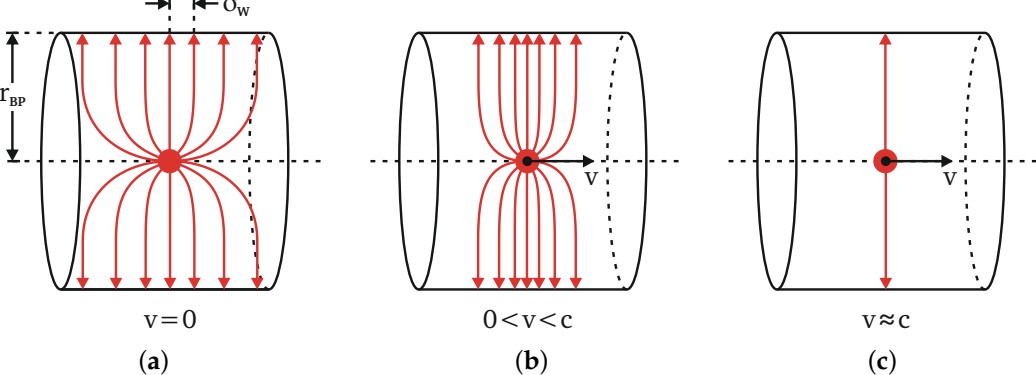

**Figure 3.** Illustration of the Lorentz contraction of the electric field lines of a single charge [5]. (**a**) Particle at rest. (**b**) Particle travelling below the speed of light. (**c**) Particle travelling close to the speed of light.

The proportionality factor between $i_{BCT}$ and $i_B$ for an FBCT in which the secondary wire makes $N$ turns around the core is equal to $N^{-1}$. The current $i_{BCT}$ is typically measured as a voltage drop $v_{BCT}$ across a load impedance $Z_{BCT}$. Hence, the bunch intensity $N_B$ can be measured with the FBCT as:

$$N_B = \frac{N}{Z_{BCT}\, e_0} \int_{t_0}^{t_0+T_B} v_{BCT}(t)\, dt \tag{3}$$

In practice, the proportionality constant between the bunch intensity and the integral of the FBCT signal is established via cross-calibration with other beam-sensing instruments. In the LHC, the reference values for total beam intensity are obtained from DC beam current transformers (DC BCTs) [6].

Figure 4 shows a simple electrical model of the FBCT, which can be used to calculate the expected signal levels and analyse the low-frequency behaviour of the monitor.

As the sensor is essentially a current transformer in which the beam represents the primary winding, it has no response at DC and its low cutoff frequency is given by:

$$f_{l,BCT} = 2\pi \frac{R_{BCT} \parallel Z_L}{L_{BCT}} \tag{4}$$

where $R_{BCT}$ is the internal load resistance built into the FBCT, $Z_L$ represents the input impedance of the acquisition system connected in parallel ($\parallel$) to $R_{BCT}$, and $L_{BCT}$ is the inductance of the secondary winding.

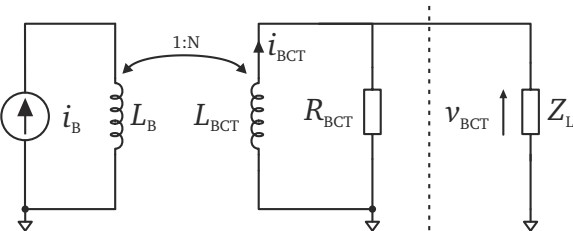

**Figure 4.** Low-frequency electrical model of the FBCT.

Due to the lack of low-frequency components, the FBCT signal is affected by the so-called baseline droop, as shown in Figure 5, which becomes more and more severe for increasing values of $f_{l,BCT}$. As the signal's baseline decreases, the droop lowers the value of the signal's integral. This leads to an error when using FBCT signals for bunch intensity monitoring. However, this error can be significantly reduced by setting $f_{l,BCT}$ such that the amount of the baseline droop over the effective integration window is negligible. For accurate bunch intensity measurements in the LHC, the low cutoff frequency of the FBCT cannot exceed 400 Hz [3]. As for the LHC FBCT $R_{BCT} = Z_L = 50\,\Omega$, which is typical for high-frequency systems, achieving a low $f_{l,BCT}$ requires sufficiently high $L_{BCT}$. For the LHC FBCT, $f_{l,BCT} \approx 200\,$Hz was obtained by winding forty secondary turns around a high-permeability nanocrystalline magnetic core [3].

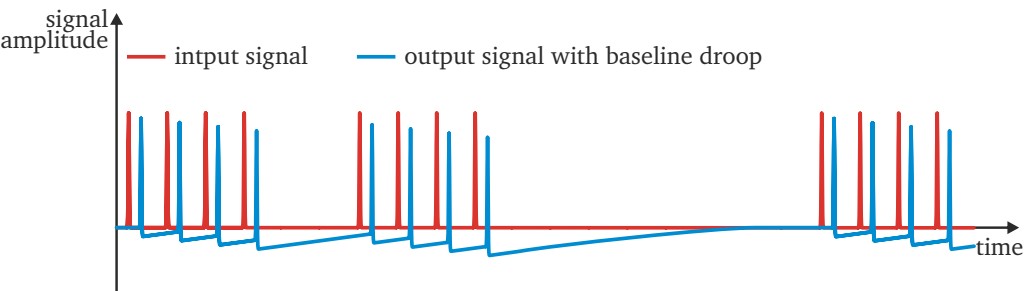

**Figure 5.** Baseline droop in the output signal of a sensor with no DC response.

In the first years of the LHC operation, the FBCT signals were acquired with a system based on custom analogue integrators [7]. It was later replaced by a fast-sampling system, still used operationally, implementing the numerical integration of the digitised signal [8].

Forty secondary turns terminated with $R_{BCT} \parallel Z_L = 25\,\Omega$ translate into an FBCT transimpedance $v_{BCT}/i_B$ of 625 mΩ which results in a very large output pulse amplitude of 12.5 V for even a modest nominal bunch with a peak current of 20 A. Therefore, the LHC FBCT signal has to be strongly attenuated before it can be measured by a high-speed data acquisition system.

Already in the first years of the LHC operation, some limitations of the FBCTs were observed. Most importantly, the sensors exhibited an undesired sensitivity to the transverse beam position and the bunch length. The bunch intensity measured by the FBCT would change when the beam was transversely displaced inside the vacuum chamber even though the true bunch intensity remained constant. Similar behaviour was seen when the bunch length changed. These two effects contributed to a measurement error of a few percent exceeding the original specification of $\pm 1\,\%$ for measurements averaged over 1 s [9]. The FBCTs outputs were fitted with analogue 80 MHz low-pass filters which reduced the sensitivity to the beam position and bunch length [10]. However, the effects remained measurable and were deemed a significant limitation of the LHC bunch intensity monitoring accuracy. Any modelling of these effects seemed very difficult, if at all possible, so they were studied only empirically.

The most severe limitation of the FBCT, the dependence of its output signal on the transverse beam position, was traced to the fact that the distribution of the beam's electromagnetic field was changing with respect to the beginning and the end of the core's secondary winding as the beam changed its position. Then, due to core losses, the field induced further from the winding end had a smaller contribution to the output signal than the field induced closer to the end of the winding connected to the output terminal.

The second crucial limitation of the FBCT was the size of its core, with an external diameter of 130 mm, an internal diameter of 90 mm, and a thickness of 25 mm, which required some 2.4 m of wire to make the forty-turn winding. Such a long winding is prone to parasitic capacitance between turns, which gives rise to resonances at frequencies as low as 30 MHz. This makes the FBCT inadequate to fully separate the consecutive LHC bunches spaced by 25 ns. Moreover, the forty-turn secondary winding of the large

high-permeability magnetic core forms a lossy delay line. The resulting frequency dispersion of the signals originating from different parts of the winding significantly deteriorates the FBCT's frequency response.

These two effects were convincingly demonstrated in a laboratory with a setup consisting of a one-turn loop, acting as the primary winding of the FBCT toroid, connected to a pulse generator (for time-domain measurements) or a network analyser output (for frequency-domain measurements). The output of the forty-turn secondary winding was connected to an oscilloscope or an input of the network analyser. When the one-turn primary loop was moved along the FBCT winding, large signal changes were observed in both the time and frequency domains.

## 3. Desired Characteristics of a New LHC Bunch Intensity Sensor

The FBCT imperfections were understood as coming from the monitor itself, rendering them very improbable to effectively overcome. Therefore, CERN decided to launch a fully in-house development aiming to find a solution that would eventually overcome the limitations and replace the LHC FBCTs.

In order to limit the changes to other LHC components, it was decided that the new sensor must ensure mechanical compatibility with the existing vacuum chamber and the dielectric insert over which the FBCTs were installed. These requirements constrained the mechanical dimensions of the new sensor to a minimum internal diameter of 84 mm, an external diameter smaller than 300 mm, and a total length of 40–290 mm.

The performance specifications have also been revised compared to the original LHC requirement of ±1 % accuracy. For the new monitor, the combined bunch intensity measurement error should not exceed 0.1% for measurements averaged over 1 s. The sensor's bandwidth should span from the low cutoff frequency $f_l < 640$ Hz up to the high cutoff frequency $f_h > 59$ MHz.

The output signal of the new sensor must be adapted to acquisition electronics based on both analogue integrators and fast sampling. The signal amplitude at the input of the data acquisition electronics should not exceed ±1.2 V and the duration of a pulse generated by a single LHC bunch should be less than 22 ns.

## 4. LHC Wall Current Transformer

To overcome the inherent FBCT design flaws, its replacement cannot employ a large magnetic core. Therefore, the standard solution with a single toroid around the beam vacuum chamber was immediately rejected. It is then evident that the new sensor cannot directly sense the beam current itself but should rather rely on the beam image current. To remain insensitive to the beam position changes, the image current must be probed at several radial positions. The corresponding individual signals must be summed with a power combiner to ensure the linearity of the signal superposition and the independence of the beam position. Finally, to maximise the sensor's bandwidth, its magnetic cores should be as small as possible and contain only a few secondary turns wound with a short wire.

With the above considerations in mind, the new sensor, named by the authors the wall current transformer (WCT), was conceived. It does not couple directly to the bunch current but rather to the current induced by it on the conductive walls of the vacuum chamber [5]. Following Gauss's law, the beam charge $Q_B$ induces an equal charge of the opposite sign $Q_W$ on the inner walls of the accelerator's vacuum chamber [11]. This phenomenon, as illustrated in Figure 6, is typically referred to as the image or wall charge.

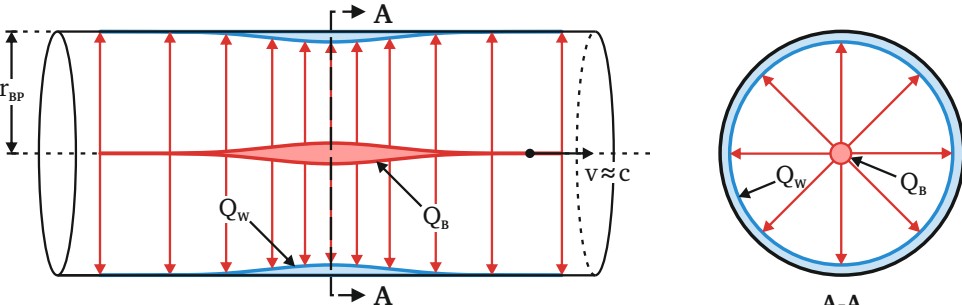

**Figure 6.** Image charge induced on the wall of a vacuum chamber by an ultra-relativistic beam [5].

Due to the Lorentz contraction of the field lines, the longitudinal distribution of the image charge for ultra-relativistic particle beams travelling very close to the speed of light is the same as that of the bunch charge. Both charges move together, which gives rise to an electric current flowing on the walls of the vacuum chamber. This current is interchangeably referred to as the image current or the wall current $i_W$. However, as the induced image charge is of the opposite sign to the bunch charge, both currents also have the same magnitude but the opposite polarity:

$$i_W(t) = -i_B(t) \tag{5}$$

Instead of using one large high-permeability toroidal magnetic core, the WCT uses eight small toroidal cores made of a nanocrystalline magnetic material. The cores are attached to a printed circuit board (PCB) and are evenly distributed around the dielectric insert embedded into the vacuum chamber. Since, for the wall current, the dielectric insert constitutes a high-impedance discontinuity, the current is forced to flow via conductive screws going through the centre of each toroid. The cores serve as RF current transformers with their primary windings formed by the screws. A thin wire is wound around each core as the secondary winding. An additional wire making a single turn around each core acts as a calibration winding allowing for a reference calibration current to be sent to the sensor.

Devices using a similar beam-coupling method were realised in the past [12–14] but they were all designed for measuring the transverse beam position at relatively low frequencies. The WCT developed by the authors is optimised for the precise measurement of the intensity of the LHC short bunches.

Figure 7 illustrates the WCT principle of operation. When the wall current reaches the monitor, all the components above the low cutoff frequency flow through the conductive screws and are sensed by the RF transformers. To avoid radiating the beam's electromagnetic field towards other accelerator components, the sensor is enclosed in a conductive housing through which low-frequency image current can flow. The WCT housing is filled with high-permeability ferrite cores which magnetically load the housing to increase its inductance. This lowers the frequency at which the image current starts flowing through the conductive screws. Both sides of the dielectric insert are connected with an additional RF bypass circuit composed of capacitors and resistors to provide a well-defined path for the high-frequency image current components. Such high frequencies are beyond the operational bandwidth of the WCT and should therefore be bypassed to avoid an excitation of the parasitic RF cavity formed by the WCT housing, which would deteriorate the longitudinal beam-coupling impedance of the WCT.

An equivalent electric model of the WCT is shown in Figure 8, which illustrates its principle of operation and can be used to calculate the most important parameters.

The image current $i_I$ has an equal magnitude but the opposite polarity to the beam current $i_B$. Inside the WCT, $i_I$ is divided into three constituents:

- The low-frequency $i_{LF}$ flowing through the housing with inductance $L_{LF}$ defined by the housing geometry and the ferrite permeability;
- The very-high-frequency $i_{RF}$ flowing through the RF bypass with capacitance $C_{RF}$, resistance $R_{RF}$, and some parasitic inductance $L_{RF}$;
- The intermediate-frequency $i_W$ flowing through the screws with inductance $L_W$ and resistance $R_W$ defined by the RF transformer.

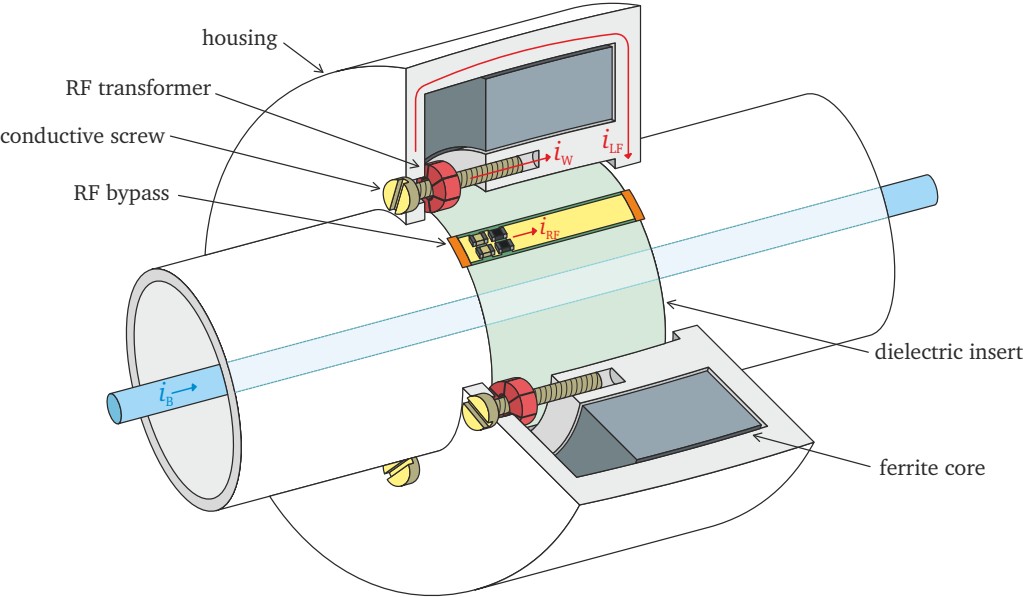

**Figure 7.** Principle of operation of the wall current transformer (WCT) [5].

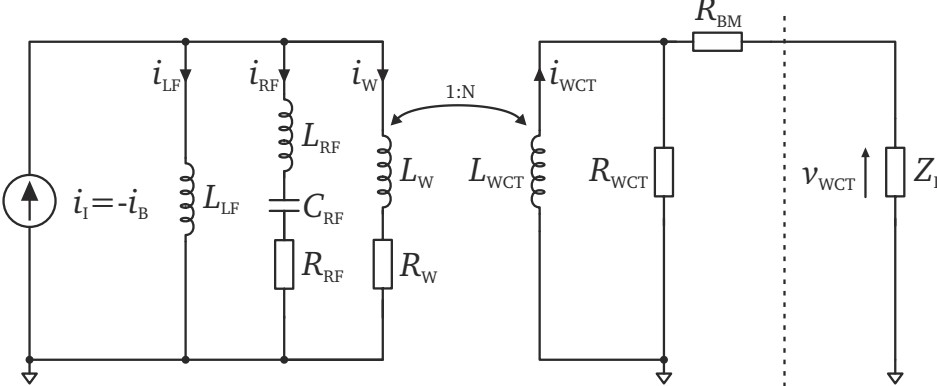

**Figure 8.** Electrical model of the WCT [5].

As in any current transformer, $i_W$ induces a current $i_{WCT}$ on the RF transformer's secondary side represented by $L_{WCT}$. This rises a voltage $v_{WCT}$ across the load resistor $R_{WCT}$. To match the WCT output to the standard $50\,\Omega$ characteristic impedance of high-frequency transmission lines, the sensor includes a back-matching resistance $R_{BM}$ which sets the monitor's output impedance to $50\,\Omega$. The WCT output signal can then be measured by an acquisition system, represented by $Z_L$.

For frequencies at which $i_W \approx -i_B$, the WCT output voltage $v_{WCT}$ is given by:

$$v_{WCT} = -\frac{1}{N}\frac{R_{WCT}Z_L}{R_{WCT} + R_{BM} + Z_L}i_B \tag{6}$$

After matching the source and load impedances by selecting resistors such that $R_{WCT} + R_{BM} = Z_L$, the above equation becomes:

$$v_{WCT} = -\frac{R_{WCT}}{2N} i_B = -Z_{WCT} i_B \tag{7}$$

As $v_{WCT}$ is proportional to the instantaneous beam current, the WCT can measure bunch intensity. The factor $Z_{WCT}$ is called the transimpedance and it is helpful for comparing various bunch intensity monitors. In practice, $Z_{WCT}$ is a function of frequency.

The frequency range at which Equation (7) is valid is determined by the sensor's bandwidth defined by its low and high cutoff frequencies $f_L$ and $f_H$, respectively.

Analysis of the circuit shown in Figure 8, assuming that $L_{LF} \gg L_W$ on the primary side and that $R_{WCT} \ll R_{BW} + Z_L$ on the secondary side, leads to a simple equation for an approximate value of $f_L$:

$$f_L \approx \frac{1}{2\pi} \left( \frac{R_W}{L_{LF}} + \frac{R_{WCT}}{L_{WCT}} \right) \tag{8}$$

As $R_W \approx R_{WCT}/N^2$ and $L_{WCT} \propto N^2$, increasing the number of secondary turns $N$ quickly decreases $f_L$. However, as shown before in Equation (7), a higher $N$ also reduces the WCT output voltage, and therefore a compromise between the cutoff frequency and the sensitivity must be found.

The precise modelling of the WCT on the high-frequency side is difficult, mostly due to many parasitic capacitive effects that can be collectively referred to as the interwinding capacitance. Nevertheless, from the circuit shown in Figure 8, assuming that, at high frequencies, the screw's impedance is dominated by $L_W$ and that the RF bypass capacitance $C_{RF}$ can be neglected, the following equation can be drawn up:

$$f_H \approx \frac{R_{RF}}{2\pi \sqrt{L_W^2 - L_{RF}^2}} \tag{9}$$

Deriving a corresponding formula for the FBCT is not feasible, mostly due to the fact that its high frequency behaviour is not well defined as it is not evident how the image current traverses the sensor. The FBCT could also be equipped with an RF bypass, similar to the one used in the WCT, which would limit the sensor's longitudinal impedance presented to the beam. Nevertheless, such an addition to the FBCT would not solve its other fundamental limitations addressed by the WCT design.

The circuit shown in Figure 8 does not implicitly account for the eight RF transformers installed in parallel inside the WCT. The components only represent the effective values of a simplified equivalent circuit rather than physical components installed in the sensor.

At the heart of the WCT, there are eight RF transformers based on Vacuumschmelze T60006-L2009-W914 toroids made of nanocrystalline iron-based VITROPERM 500 F material. They were selected due to their high inductance factor $A_L = 25.5\,\mu H$ at 10 kHz and a sufficiently small size. The cores, after stripping them of their protective plastic casing, have dimensions of only 6.5 mm (inner diameter) by 9.9 mm (outer diameter) by 4.8 mm (height).

Each core is wound with $N = 10$ secondary turns and loaded with $5\,\Omega$. With eight transformers in parallel, this translates into an effective $R_{WCT} = 625\,m\Omega$. From Equation (7), the WCT transimpedance is $Z_{WCT} = 31.25\,m\Omega$ which is 20 times smaller than that of the FBCT. With the typical LHC bunch currents, before any filtering or attenuation, the WCT output would have an amplitude of 31 mV for a pilot bunch and 625 mV for a nominal bunch. Such levels are perfectly appropriate for typical front–end signal conditioning electronics.

To improve the low-frequency behaviour of the WCT, the housing was filled with Ceramic Magnetics CMD5005 nickel–zinc machined ferrite cores with relative permeability $\mu_r = 2100$ up to 600 kHz and overall dimensions of 176.8 mm (outer diameter), 117.2 mm (inner diameter), and 60 mm (length). With such a high-permeability core inside, the WCT housing has an inductance of $L_{LF} = 10\,\mu H$.

The effective secondary-side inductance $L_{WCT} = 320\,\mu H$ is formed by eight parallel cores with a ten-turn winding each. Together with the aforementioned housing inductance $L_{LF}$ and the secondary-side resistance $R_{WCT}$, it is possible to use Equation (8) to calculate the low cutoff frequency of the WCT as $f_L = 410\,Hz$.

An internal RF bypass made from two flexible PCBs controls the WCT's high-frequency behaviour. The bypass consists of series resistors and capacitors with effective values of $R_{RF} = 1.67\,\Omega$ and $C_{RF} = 60\,nF$. The capacitance decouples the bypass at low frequencies, forcing the current to flow through the conductive screws instead. At very high frequencies, the bypass and the screws form a current divider. The screws' impedance is dominated by their self-inductance of approximately $L_W = 250\,pH$. The RF bypass' parasitic inductance was conservatively estimated as $L_{RF} < 50\,pH$. Therefore, the theoretical high cutoff frequency of the WCT calculated from Equation (9) is at least 1.1 GHz.

To produce a single WCT output, the signals generated by the eight RF transformers are added up through a network of passive power combiners. The outputs of two adjacent transformers are directly averaged on the internal WCT PCBs. Each 5 Ω secondary-side load is followed by a series 95 Ω back-matching resistor to set the source impedance to 100 Ω. Short transmission lines of the same characteristic impedance merge pairs of transformer outputs into four intermediate WCT outputs, thus becoming 50 Ω sources. The intermediate outputs can then be summed with three external resistive power combiners producing a single common WCT output. To mitigate the sensor's sensitivity to the transverse beam position, the internal WCT resistors are matched to within 0.01%.

The passive four-way combiner used at the WCT output generates a thermal noise with a spectral density of $2\,nV/\sqrt{Hz}$. Within the theoretical 1.1 GHz bandwidth of the WCT, this corresponds to the root-mean-square (RMS) noise of 66 μV. Numerical simulations show that the RMS value of an expected WCT output signal shape calculated over a 25 ns window is equal to nearly 15% of the peak voltage. Therefore, the RMS WCT output signal level is 5 mV for a pilot bunch and 94 mV for a nominal bunch. The resulting signal-to-noise ratio (SNR) is 37 dB for a pilot bunch and 63 dB for a nominal bunch which is equivalent to a non-averaged measurement error of 1.3% and 0.07%, respectively. However, as the LHC bunch intensity measurements are averaged over 1 s, i.e., over 11,245 samples, the effective SNR and measurement error are better by some 40 dB, i.e., two orders of magnitude.

Besides the secondary winding generating beam-related signal, each RF transformer of the WCT has an additional one-turn winding which can accept external calibration signals. The WCT is optimised for calibration with long current pulses, the amplitude of which can be precisely measured. The input impedance of its calibration port is 10 Ω which, for a 1 A current, generates a modest voltage of 10 V. To decouple the calibration winding from the high-frequency current flowing through the conductive screw, the calibration signal path includes several ferrite beads with a high impedance at RF frequencies.

Figure 9 shows a schematic of the internal LHC PCBs together with the external combiner network. The schematic does not account for the sensor's conductive screws.

Figure 10 shows the WCT's internal PCBs and some details of RF transformers. All boards were assembled by hand using manually matched resistors. The PCBs only use the top layer for signal routing and the impedance of all traces is controlled. The RF transformers were carefully wound by hand and fixed to the PCB with a small amount of acrylic adhesive which was also used to secure the windings to the core. The four intermediate signal outputs, as well as the calibration inputs, use standard SMA connectors (not visible in the photographs).

A noteworthy feature of the WCT mechanical design is that all its parts are cut in half to allow the assembly and disassembly of the sensor around a closed LHC vacuum chamber. Such a solution was chosen to make it possible to install and remove the first prototype WCT without uninstalling the original FBCT from the accelerator. Instead, the FBCT could be simply slid away from the dielectric insert along the vacuum chamber. Figure 11 show both sensors installed in the LHC side-by-side. However, in a general case,

when retrofitting is not required, the WCT mechanical design can be simplified if its parts are not cut in half.

Further technical details about the WCT and a thorough derivation of the sensor's electrical model can be found in [5].

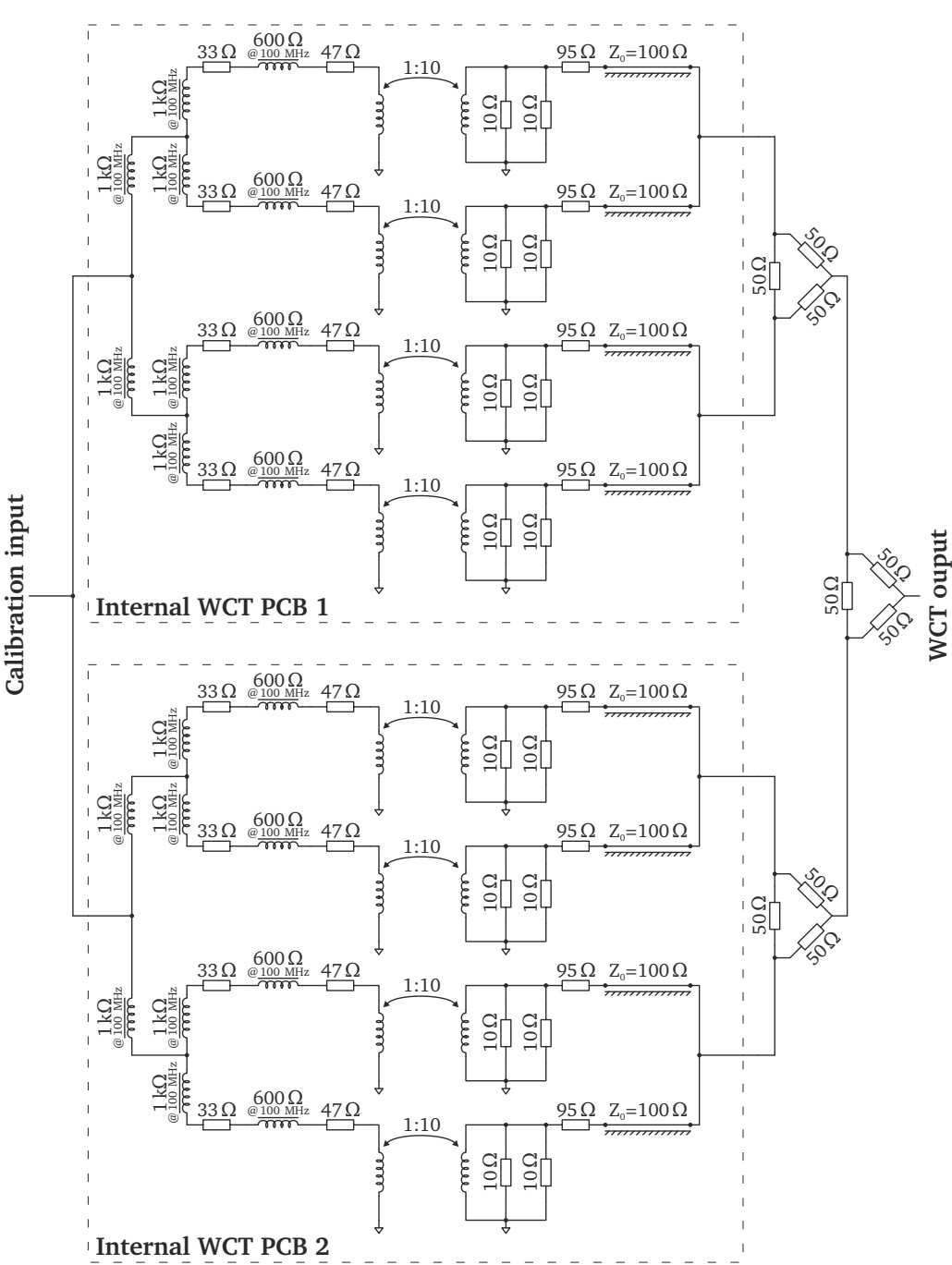

**Figure 9.** Schematic of the internal WCT PCBs [5].

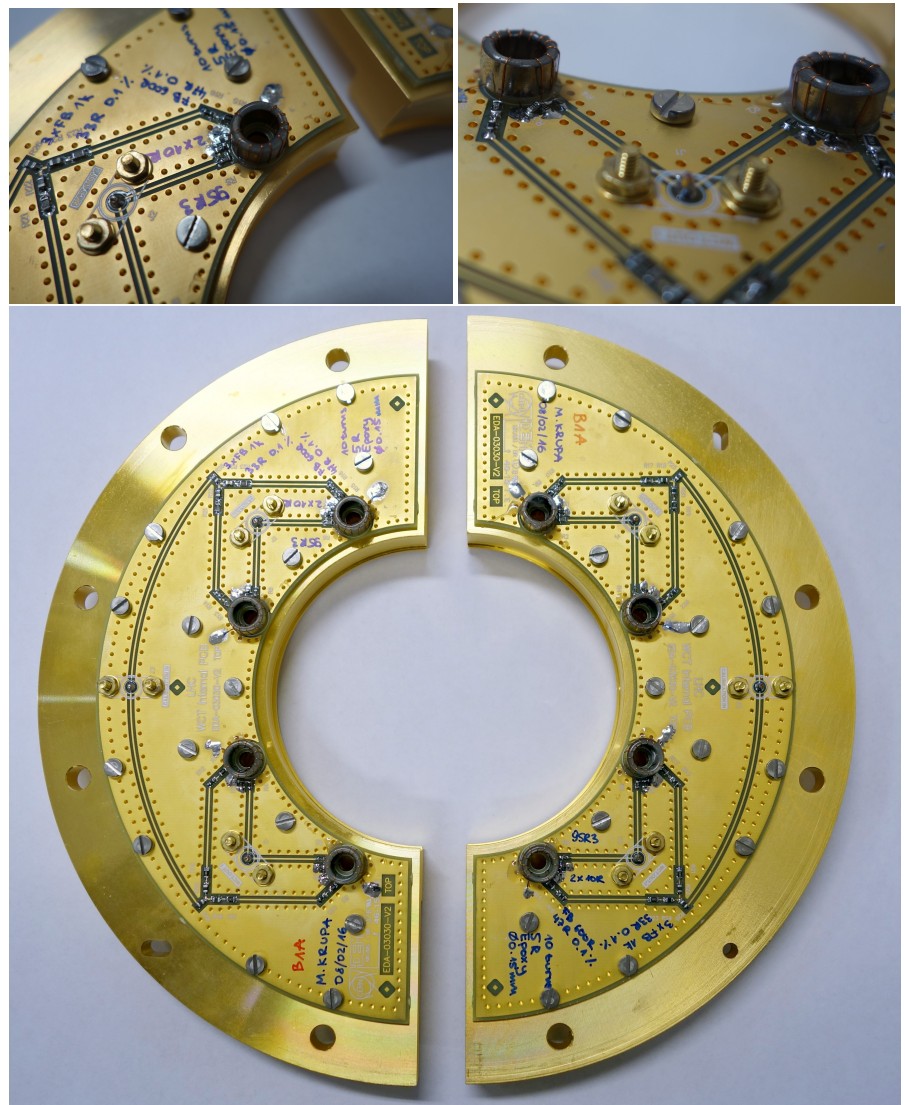

**Figure 10.** Internal WCT PCBs [5].

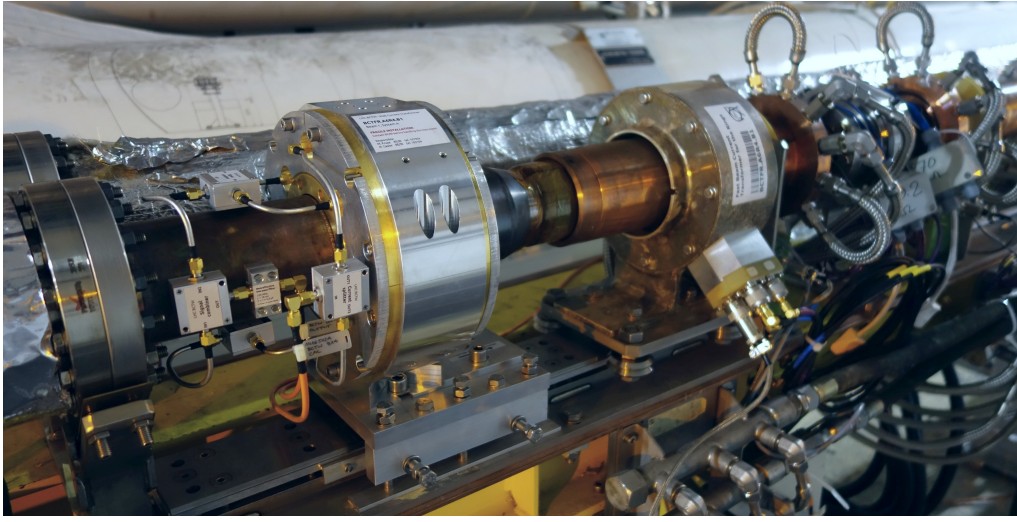

**Figure 11.** The new WCT (left) installed next to a displaced FBCT (right) [5].

Besides the sensor itself, a complete set of custom analogue front–end electronics has been designed and deployed for the LHC WCT with their functional diagram shown in Figure 12. The WCT output signal's bandwidth is first reduced by a non-reflective linear-phase low-pass filter (LPF) directly located after the final signal combiner. The signal then is sent over a short run of low-loss coaxial cable to the head amplifier located closer to the accelerator tunnel floor. The head amplifier provides two copies of the signal with 20 dB amplitude difference, foreseen for low- and high-intensity bunches. The signals are then sent through about 20 m of low-loss coaxial cable to a nearby technical gallery where the remaining electronics are well shielded from ionising radiation present during the LHC operation. A common-mode (CM) choke suppresses any interference picked up on the cable. The signal bandwidth is further reduced by another LPF and the signal is boosted by a distribution amplifier which makes four copies of each signal. Each output of the distribution amplifier is equipped with an LPF and an attenuator to adapt the signal to a given acquisition system. The expected signal levels along the WCT signal path are listed in Table 1. The amplifiers lose linearity with outputs exceeding 2.7 V and saturate at around 3.8 V; therefore, the high gain channel cannot be used for observing high-intensity bunches.

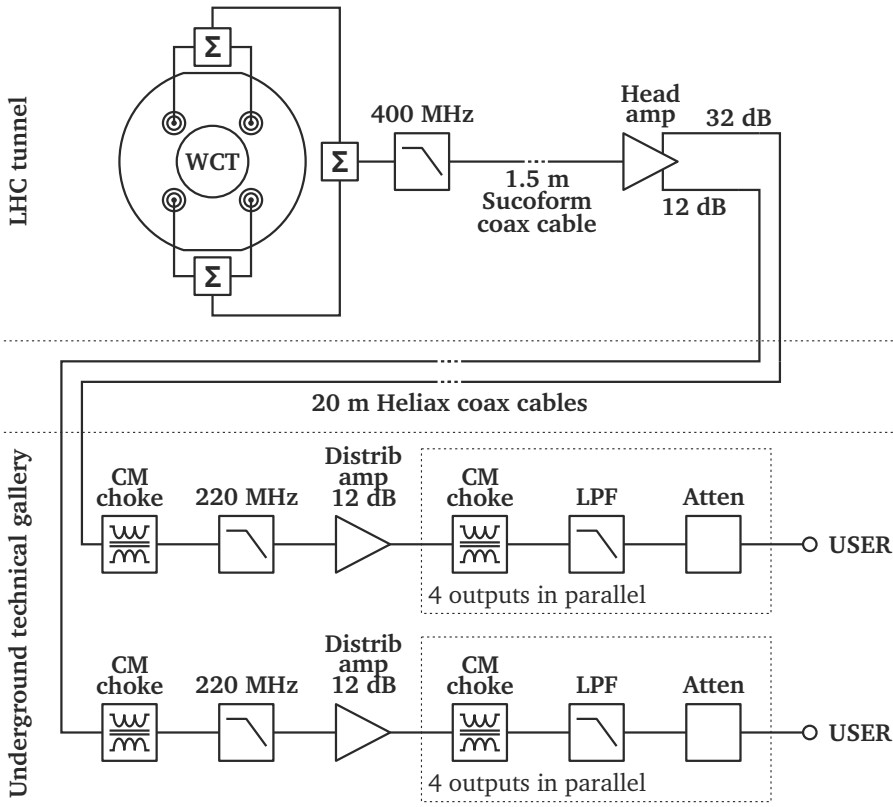

**Figure 12.** WCT front–end electronics diagram [5].

The WCT's noise performance is dominated by the noise of the head amplifier which has an RMS value of 0.4 mV and 2.5 mV at the output of the "low" and "high gain" channels, respectively. The RMS value of a WCT head amplifier output signal calculated over a 25 ns window equals approximately 25% of the peak voltage. Therefore, from the estimated signal levels listed in Table 1, it is possible to calculate an SNR of 34 dB for a pilot bunch measured with the "high gain" channel and 57 dB for a nominal bunch measured with the "low gain" channel. Similarly to the SNR directly calculated at the sensor output, the effective SNR for measurements averaged over a 1 s window is better by two orders of magnitude. Hence, the noise-related measurement error is well below the required 0.1%.

Further technical details concerning WCT electronics can be found in [5].

**Table 1.** Signal levels of the WCT and its front–end electronics with typical LHC bunches.

|  |  | **Pilot** | **Nominal** |
|---|---|---|---|
| **Bunch** | Bunch intensity (charges) | $5 \times 10^9$ | $1.2 \times 10^{11}$ |
|  | Bunch length (ns) | 1.2 | 1.2 |
|  | Peak bunch current (A) | 1.1 | 25.5 |
| **WCT output amplitude (mV)** | In 1.1 GHz BW | 29 | 694 |
|  | In 400 MHz BW | 20 | 460 |
| **Head amplifier amplitude (mV)** | Low gain (in 220 MHz BW) | 51 | 1184 |
|  | High gain (in 220 MHz BW) | 517 | Saturated |
| **Distribution amplifier amplitude (mV)** | Low gain (in 70 MHz BW) | 70 | 1621 |
|  | High gain (in 70 MHz BW) | 708 | Saturated |

## 5. Results

The WCT and FBCT were compared through extensive laboratory tests followed by beam measurements in the LHC. The first prototype WCT was installed in the place originally occupied by the FBCT, which was temporarily slid away from its dielectric insert. As the WCT can be assembled and disassembled around a closed vacuum chamber, such an approach allowed us to thoroughly test the new detector with the real LHC beam, leaving the option of a quick return to the original sensor in case any problems were discovered.

Figure 13 shows the FBCT and WCT amplitude-normalised time response to a real nominal LHC bunch (bunch intensity of $1.1 \times 10^{11}$ protons, bunch length of 1.2 ns, beam energy of 450 GeV) measured with a Teledyne Lecroy HDO6104 12-bit oscilloscope (procured from Teledyne Lecroy SA, Vernier, Switzerland) with 1 GHz analogue bandwidth sampling at an equivalent rate of 125 GS/s installed close to the sensor. To reduce the noise level, 100 consecutive acquisitions were averaged. The FBCT was equipped with its standard 80 MHz LPF, while the WCT was measured in two configurations: unfiltered full bandwidth and through a 120 MHz non-reflective LPF.

The FBCT produces a pulse of about 12 ns followed by a tail lasting some 40 ns. The FBCT response clearly extends over 25 ns, potentially overlapping with a subsequent bunch.

The unfiltered WCT response is short enough to be easily shaped with external low-pass filters. After passing through a 120 MHz LPF, the signal almost fills the entirety of the available 25 ns bunch window, leaving approximately 2 ns of baseline following the pulse. This proves that the WCT signal allows the individual LHC bunches to be clearly distinguished and that there is no signal leakage to subsequent bunch slots.

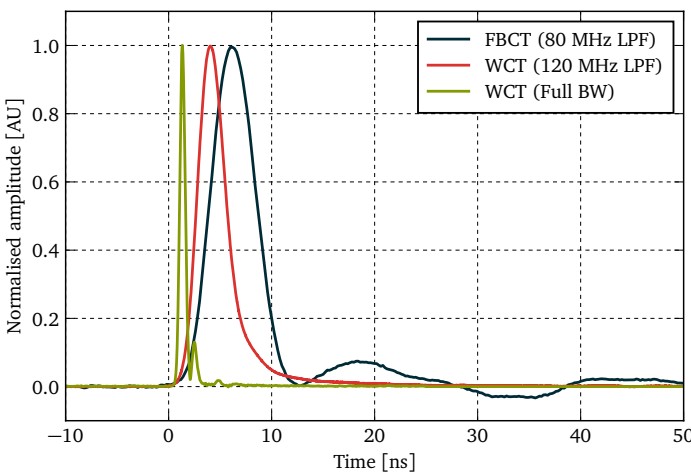

**Figure 13.** Amplitude-normalised time response of the WCT and the FBCT to a nominal LHC bunch [5].

Figure 14 plots the absolute signal levels measured in the same way at the output of the WCT distribution amplifier for typical LHC bunches. For comparison, the right plot also includes a response of the WCT to the nominal bunch measured directly at the sensor's output. A pilot bunch with an intensity of $5.8 \times 10^9$ protons measured through the "high gain" channel generates a pulse with a 702 mV amplitude. This can be translated into a sensitivity of 121 mV/$10^9$ ppb. The "low gain" channel outputs peaks at 1363 mV for a nominal bunch with an intensity of $1.2 \times 10^{11}$ protons yielding a sensitivity of 11 mV/$10^9$ ppb.

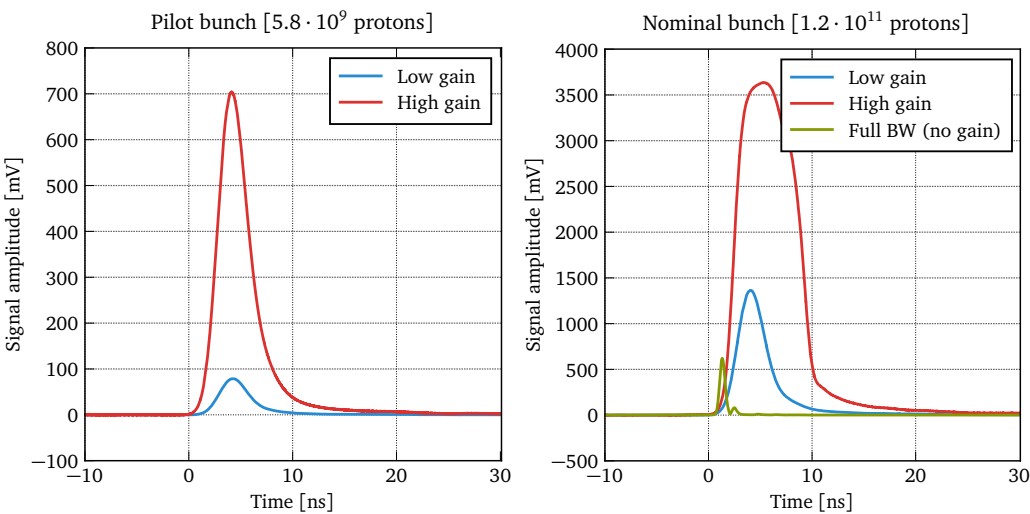

**Figure 14.** WCT response to pilot and nominal LHC bunches measured at the output of the distribution amplifier [5].

Figure 15 shows the frequency response of the FBCT and the WCT in the range of 1–2000 MHz. The sensors were measured without any additional filtering ("full BW") and with their typical external LPF. The measurements were performed with an Agilent Technologies E5071C Vector Network Analyzer (VNA) (procured from Agilent Technologies Schweiz AG, Basel, Switzerland) on a custom-built laboratory coaxial test setup, as illustrated in Figure 16. The data were normalised to equal 0 dB at 1 MHz.

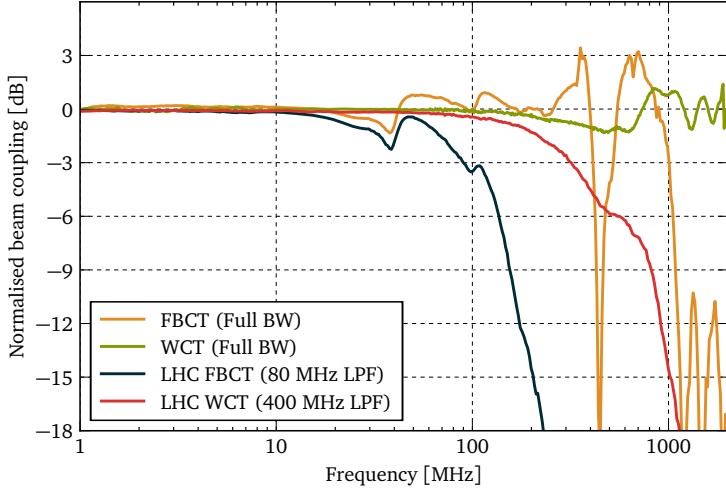

**Figure 15.** WCT and FBCT frequency response measured on a laboratory coaxial test bench [5].

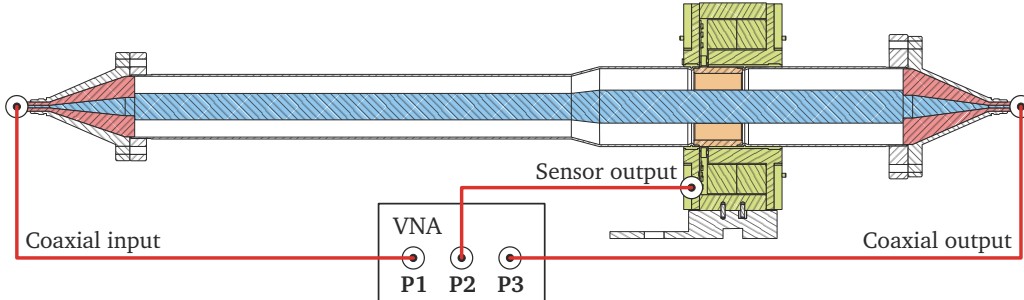

**Figure 16.** Connection diagram for frequency domain measurements [5].

The non-filtered FBCT has a high cutoff frequency close to 1 GHz, but the sensor's imperfections start being clearly visible already above 30 MHz with a very distinct resonance at 450 MHz. The FBCT's 80 MHz LPF strongly attenuates the high-frequency resonance, but the magnitude fluctuations in the range of 30–100 MHz remain visible.

The frequency–domain measurements of the WCT further substantiate its excellent time–domain performance. The frequency response remains flat within $\pm 1$ dB over the entire measurement range. Even though some mild fluctuation can be seen above 700 MHz, the LHC bunches carry relatively little power at such high frequencies. The standard WCT 400 MHz LPF almost completely mitigates the high-frequency imperfections without compromising the sensor's time–domain response.

The laboratory coaxial test setup was also used to quantify the longitudinal beam-coupling impedance of the FBCT and the WCT. The beam's electromagnetic field unavoidably interacts with any component installed on the vacuum line inside which the beam travels. For components made from certain material and with certain geometries, the amount of electromagnetic energy extracted from the beam might be significant and lead to heat generation and other detrimental consequences. These effects are typically modelled as an additional impedance that the component exerts on the beam, and hence are referred to as the longitudinal beam-coupling impedance [15].

The longitudinal beam-coupling impedance of the FBCT and the WCT, as shown in Figure 17, was measured using the traditional stretched-wire technique [16]. With its well-defined paths for high-frequency currents, the WCT impedance remains below 6 $\Omega$ over the entire tested frequency range. On the other hand, the FBCT impedance is significantly higher and features two strong peaks at 380 MHz and 1.1 GHz. Even though both monitors are very minor contributors to the overall LHC impedance budget, the WCT qualitatively demonstrates another improvement over the FBCT design.

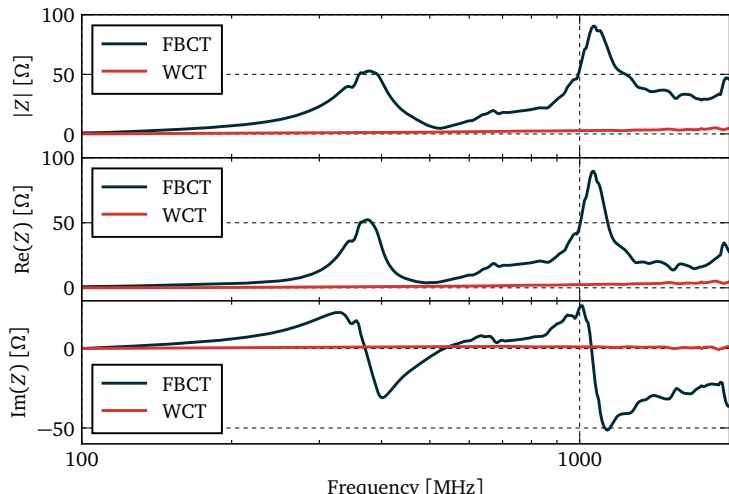

**Figure 17.** WCT and FBCT beam-coupling impedance measured on a laboratory coaxial test bench [5].

The sensitivity of the WCT and the FBCT to the transverse beam position and the bunch length were quantified during dedicated measurement sessions with the LHC beams. For the former, a beam consisting of five nominal bunches with an intensity of approximately $1.05 \times 10^{11}$ ppb each was displaced in a series of steps, as plotted in Figure 18. The top plot shows the total beam intensity (i.e., the sum of five individual bunch intensities) measured by the FBCT and the WCT. The bottom plot displays by how much the beam was displaced in the horizontal (H) and vertical (V) plane from its average orbit at the location of the FBCT and the WCT. These values were calculated by interpolating the measurements of the closest upstream and downstream beam position monitors.

The total beam intensity measured by both sensors steadily decreased by about 0.5% over the data collection period. This decay is a natural phenomenon as the bunches lose some of their particles over time. However, the FBCT measurements are also visibly correlated with the transverse beam position and they vary by 0.5–0.8% mm$^{-1}$ depending on the plane in which the beam was moved. Conversely, the WCT displayed no sensitivity to the transverse beam position down to the detection limit of 0.005% mm$^{-1}$. The measurements prove that the outputs of the WCT's eight internal RF transformers are combined with an excellent symmetry.

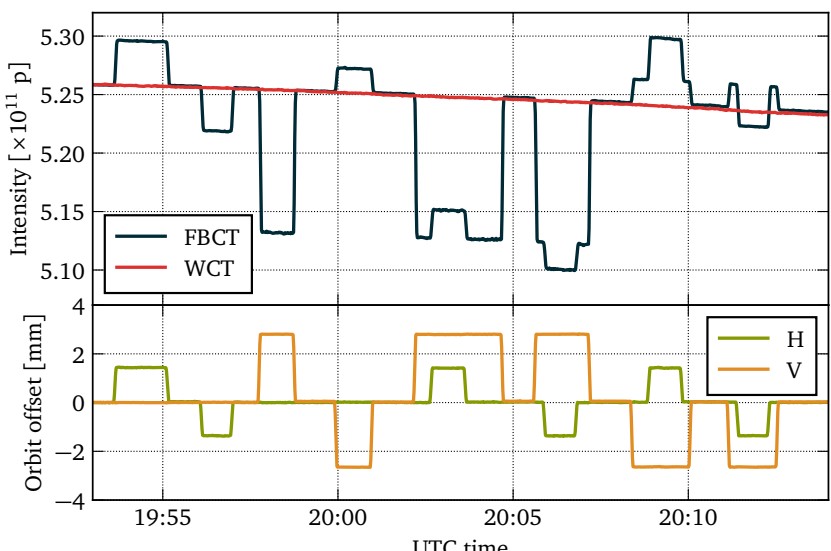

**Figure 18.** Sensitivity of the WCT and FBCT to the transverse beam position measured with the LHC beam [5].

The sensitivity of both sensors to the bunch length was tested with a beam consisting of 13 nominal bunches with a total intensity of around $13.1 \times 10^{11}$ protons. The bunches were shortened and lengthened by adjusting the amplitude of the sinusoidal electric field applied to the bunches by the LHC RF cavities. Figure 19 shows the total beam intensity measured by the FBCT and the WCT when the bunch length was intentionally changed. The average bunch length, as shown in the bottom plot, was recorded by the beam quality monitor (BQM). Besides the natural slow beam intensity decay, both monitors also recorded a sharp drop of approximately 0.02% when the bunch length was quickly increased by 200 ps. However, when the bunch length was just as quickly decreased by 200 ps, a minute later, the readings of both sensors remained stable. Therefore, the observed drop is considered to be an observation of a true beam loss rather than a proof of bunch length sensitivity of either monitor. Overall, neither the FBCT nor the WCT displayed any sensitivity to the bunch length within the detection limit of 0.2% ns$^{-1}$. Such a result obtained for the FBCT is better than what had been previously published [10]. It is assumed that the apparent elimination of the FBCT's bunch length sensitivity might be caused by the 80 MHz low-pass filter installed on the FBCT output, which stretches the signal pulses to the extent that their shapes do not change significantly with the bunch length variations.

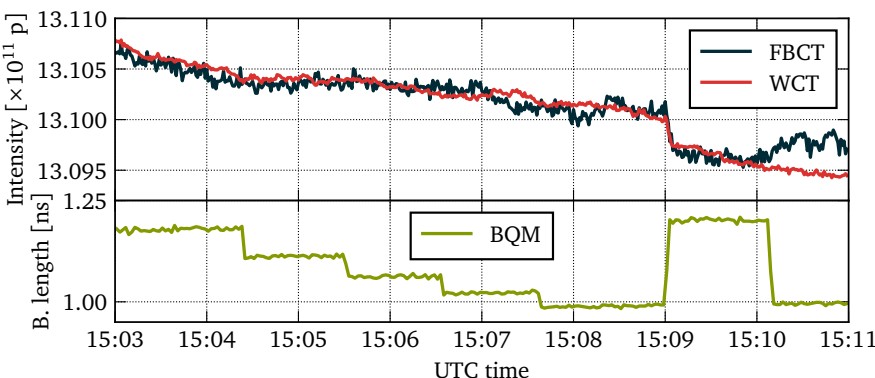

**Figure 19.** Sensitivity of the WCT and FBCT to the bunch length [5].

A practical example demonstrating the difference in performance of the WCT and the FBCT under real operational conditions are beam chromaticity measurements. Chromaticity links the beam's transverse oscillation frequency and its momentum and is one of the most fundamental parameters to monitor and control in a circular accelerator such as the LHC [17]. Chromaticity is typically measured by modulating the RF frequency used to accelerate the beam while keeping a constant field in the bending magnets. This results in a sinusoidal modulation of the horizontal beam position. Figure 20 shows an example of beam intensity measurements performed by the WCT and the FBCT of a beam consisting of eight pilot bunches at an energy of 450 MeV as the RF frequency is being modulated for beam chromaticity measurements. The FBCT readings are evidently correlated with the RF frequency while the WCT behaves as expected from a good beam intensity sensor.

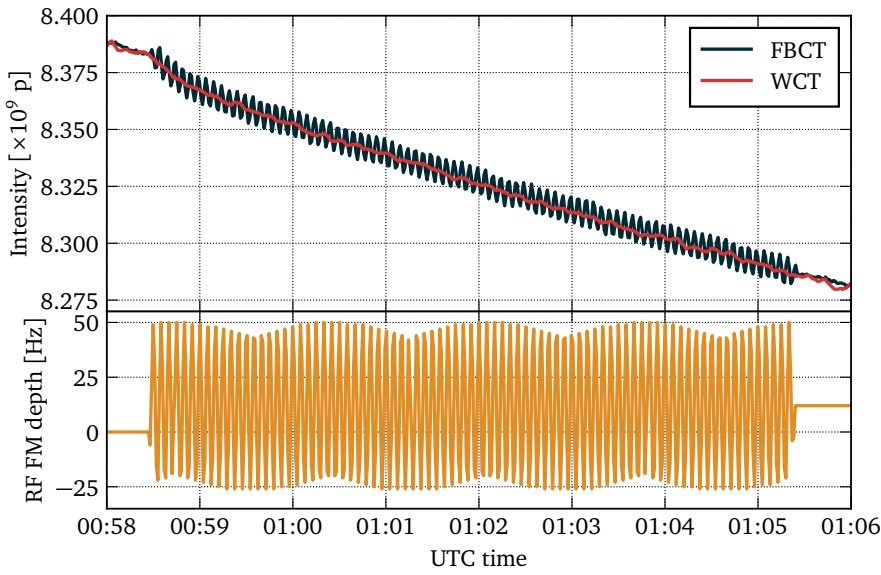

**Figure 20.** Beam intensity measured by the FBCT and the WCT during RF Frequency Modulation (FM) for beam chromaticity measurements [5].

Another situation in which the beam position can drastically change during operation is the calibration process of one of the beam profile monitors which requires displacing the beam by several millimetres [18]. This procedure is carried out several times each year with Figure 21 showing one such period. The intensity of a beam consisting of a single nominal bunch was measured by the WCT and the FBCT (top plot), while the change in the beam position was recorded by nearby beam position monitors. Once again, WCT's insensitivity to the beam position results in much more accurate intensity measurements than those provided by the FBCT.

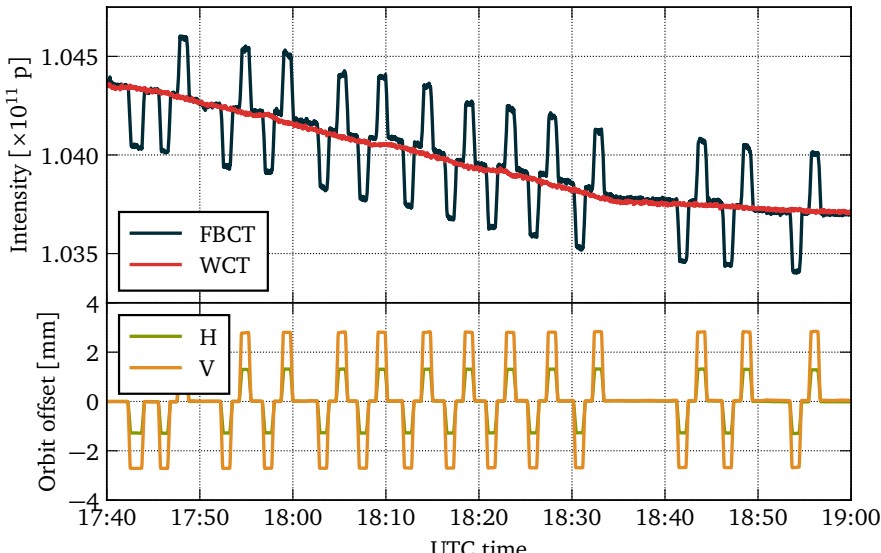

**Figure 21.** Beam intensity measured by the FBCT and the WCT during beam profile monitor calibration procedure [5].

The prototype WCT either matched or exceeded the FBCT performance in every laboratory and beam test. Based on these results, the WCT became the new reference LHC bunch intensity sensor. The four FBCTs originally installed in the LHC were removed and replaced by four WCTs. Moreover, an additional WCT of the same design was installed in the second largest accelerator at CERN, namely the Super Proton Synchrotron (SPS).

## 6. Conclusions

The developed WCT has successfully addressed the main performance limitations of the original LHC bunch intensity monitor, the FBCT. Due to its relatively large magnetic toroid, the FBCT's frequency response is insufficient to cleanly distinguish the consecutive LHC bunches. Moreover, the inherent asymmetry of the core secondary winding with respect to the particle beam results in a beam position dependence of the FBCT output signal. The WCT design addresses the former issue by using much smaller RF transformers producing a better frequency response. The latter limitation is overcome by employing eight transformers symmetrically distributed around the accelerator's vacuum chamber. Each of the individual transformers measures a part of the beam image current and their outputs are summed by an external RF power combiner. Consequently, the new sensor's bandwidth, transimpedance, and SNR are adequate for precise measurements of individual LHC bunches spaced by 25 ns. During dedicated measurements sessions with the LHC beam, the WCT has been conclusively demonstrated to be insensitive to the transverse beam position and the bunch length.

Table 2 summarises the most important parameters of the LHC FBCT and WCT. The notable difference of the magnetic core sizes and the secondary winding wire length are the prime factors contributing to the different high-frequency responses of the two monitors.

The WCT's mechanical design allows it to be assembled and removed without the need to vent the accelerator vacuum chamber. Thus, it was possible to install the first prototype WCT without removing the original FBCT from the LHC. Once the prototype was fully validated, the WCT became the new reference bunch intensity monitor and completely replaced the FBCT on both LHC rings. The mechanical design of the WCT can be simplified if its installation around an existing dielectric insert is not required.

Although the WCT described in this manuscript has been optimised for the measurements of the short current pulses generated by high-intensity proton bunches circulating in the LHC, the same technology can find use in other applications requiring precise non-intercepting wideband measurements of current signals. The WCT's low beam-coupling

impedance and its relatively small footprint make it a good candidate for facilities dealing with high-power beams, such as spallation sources, or where space is limited, such as medical accelerators.

**Table 2.** A summary of the most important FBCT and WCT parameters.

|  | **FBCT** | **WCT** |
|---|---|---|
| **Sensed quantity** | Beam field | Image current |
| **Number of magnetic toroids** | One large | Eight small |
| **Position component superposition** | Toroid | External RF combiner |
| **Beam position sensitivity (% mm$^{-1}$)** | 0.5–0.8 | <0.005 |
| **Bunch length sensitivity (% ns$^{-1}$)** | <0.2 | <0.2 |
| **Number of secondary turns** | 40 | 10 |
| **Secondary-side load ($\Omega$)** | 50 | $\frac{5}{8} = 0.625$ |
| **Transimpedance (m$\Omega$)** | 625 | 31.25 |
| **Low cutoff frequency (Hz)** | 200 | 410 |
| **Clean response limit (MHz)** | $\approx 30$ | $\approx 700$ |
| **Toroid dimensions (mm)** | $\approx \varnothing 130 \times \varnothing 100 \times 25$ | $\approx \varnothing 10 \times \varnothing 7 \times 5$ |
| **Secondary winding wire length (cm)** | $\approx 240$ | $\approx 8$ |

The final advantage of the WCT is that it is based on inexpensive off-the-shelf small magnetic cores, making it an interesting option for in-house developments.

**Author Contributions:** Conceptualisation, M.K. and M.G.; methodology, M.K.; validation, M.K.; formal analysis, M.K.; investigation, M.K.; resources, M.G.; data curation, M.K.; writing—original draft preparation, M.K.; writing—review and editing, M.G.; visualisation, M.K.; supervision, M.G. All authors have read and agreed to the published version of the manuscript.

**Funding:** This research received no external funding.

**Data Availability Statement:** Data available on request from the corresponding author.

**Acknowledgments:** The authors would like to thank their colleagues from the CERN Beam Instrumentation Group, especially: S. Bart Pedersen, D. Belohrad, J.J. Gras, and J. Kral (currently at DESY) for their work on the WCT acquisition systems, as well as T. Lefevre and L. Soby for their support during the development of the Wall Current Transformer.

**Conflicts of Interest:** The authors declare no conflict of interest.

## Abbreviations

The following abbreviations are used in this manuscript:

| | |
|---|---|
| CM choke | Common-mode choke |
| FBCT | Fast-beam current transformer |
| LHC | Large Hadron Collider |
| LPF | Low-pass filter |
| PCB | Printed circuit board |
| RF | Radio frequency |
| RMS | Root mean square |
| SNR | Signal-to-noise ratio |
| VNA | Vector network analyser |
| WCT | Wall current transformer |

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
