# Peer review of "A New Wall Current Transformer for Accurate Beam Intensity Measurements in the Large Hadron Collider"

_energies, doi:10.3390/en16217442_

Round 1
Reviewer 1 Report
Comments and Suggestions for Authors
Article describes a new bunch intensity sensor (Wall Current Transformer) used in the Large Hadron Collider. The issue therefore concerns highly specialized research equipment requiring highly advanced technology. Therefore, I have no doubt that the substantive level of the article is adequate to the rank of the scientific journal to which it is applying.
The work was written in a very clear and logical way. It contains a detailed description of the issue and the technology used so far. Then, the authors presented the concept of a new solution, presented a diagram of the system, prepared a prototype and performed tests that proved the validity of the adopted assumptions. All this was illustrated with high-quality graphics and an appropriate description. Despite my best efforts, I did not find any editorial or substantive errors in the work. The created measurement system was implemented and successfully replaced the previous generation solution, which in itself is an important confirmation of the quality of the results obtained in scientific work. All I can do is congratulate the authors.
Author Response
The authors thank the Reviewer for the time spent on reviewing our manuscript and for the very encouraging words.
Reviewer 2 Report
Comments and Suggestions for Authors
In the manuscript titled A New Wall Current Transformer for Accurate Beam Intensity Measurements in the Large Hadron Collider, I note that the most important point of the manuscript (innovation) was the replacement of the fast beam current transformer (FBCT) at the LHC with a new Wall Current Transformer (WCT) sensor, since the FBCTs exhibited several deficiencies that degraded measurement accuracy below the required level. Several comparative tests were carried out between the proposed WCT sensor and the traditional FBCTs (Normalized temporal response, Sensitivity of WCT and FBCT, etc.). Very promising results were then obtained, which led to the validation of the WCT, and it became the new reference group intensity monitor. In short, the manuscript was well prepared, presenting a good theme and of great interest to the scientific community, as well as presenting very promising results. In terms of content (engineering), the manuscript is well prepared and I have no corrections to recommend. However, I leave some formatting recommendations in order to improve the manuscript.
1st Figure 1 is illustrated before being cited in the text. I suggest that the text is cited first (line 43) and then Figure 1 is illustrated.
2nd Figure 3 is illustrated before being cited in the text. I suggest that the text is cited first (line 80) and then Figure 3 is illustrated.
3rd Figure 4 is illustrated before being cited in the text. I suggest that the text is cited first (line 90) and then Figure 4 is illustrated.
4th Figure 5 is illustrated before being cited in the text. I suggest that the text citation comes first and then comes Figure 5.
5th Figure 7 is illustrated before being cited in the text. I suggest that the text citation comes first (line 171) and then comes Figure 7.
6th Figure 12 is illustrated before being cited in the text. I suggest that the text citation comes first (line 292) and then comes Figure 12.
7th Figure 15 is illustrated before being cited in the text. I suggest that the text citation comes first and then comes Figure 15.
8th Figure 16 is illustrated before being cited in the text. I suggest that the text citation comes first (line 367) and then comes Figure 16.
9th On line 17 it is written: ``The Large Hadron Collider (LHC) at CERN is the the largest and highest-energy´´. I think there is an extra term ``the´´.
Author Response
We thank the reviewer for taking the time to review our manuscript. We reviewed the manuscript to address the reviewer's comments which improved the quality of the submitted manuscript. We provide further details to the reviewer's comments below.
Comments #1-8: Figure appearing before it is referred to in the text.
We have reorganized all figures such that they appear after they are referred to in the text.
Comment #9: On line 17 it is written: ``The Large Hadron Collider (LHC) at CERN is the the largest and highest-energy´´. I think there is an extra term ``the´´.
Thank you very much for pointing out this typo. We have corrected it in the resubmitted version of the manuscript.
Reviewer 3 Report
Comments and Suggestions for Authors
This paper is relatively original. The charts and graphs for each are beautiful. There is enough depth in this paper.
1- So the author is requested to add some more scientific introduction in the introduction section.
2- It is suggested that the authors add more experimental comparisons to fully demonstrate the superiority of the device under study.
Author Response
We thank the reviewer for taking the time to review our manuscript. We reviewed the manuscript to address the reviewer's comments which improved the quality of the submitted manuscript. We provide further details to each of the individual comments below.
Comment #1: "So the author is requested to add some more scientific introduction in the introduction section."
We have extended the introduction to include more scientific background about the LHC (lines #24-26 in the reviewed version) and the link between accurate beam (or bunch) intensity measurements and the calculation of a collider's figure of merit - its luminosity (lines #53-63 in the reviewed version).
Comment #2: It is suggested that the authors add more experimental comparisons to fully demonstrate the superiority of the device under study.
We agree that in the originally submitted version of the manuscript the WCT was compared to the FBCT only in precisely controlled test conditions. We have significantly extended the results section to include two examples of measurements taken during regular LHC operation (lines #479-497 and Figs. 20-21 in the reviewed version). Moreover, we have added a further explanation of the results observed during bunch length sensitivity studies (lines #474-478 in the reviewed version). We have also added a description of previous laboratory studies conducted by one of the authors on FBCT's sensitivity to the beam position and bunch length (lines #154-176 in the reviewed version).
Reviewer 4 Report
Comments and Suggestions for Authors
1. It is recommended that the authors use the equivalent models shown in Figure 4 and Figure 8 to explain why the proposed WCT is superior to the traditional FBCT in electrical characteristics.
2. The author should sort out all the comparison items and corresponding quantitative data between the proposed WCT and traditional FBCT, and make a table to understand the differences in characteristics.
3. It is recommended that the author draw the test platform connecting WCT, FBCT, and related test instruments (including equipment models) in section 5 and list all test conditions.
Author Response
We thank the reviewer for taking the time to review our manuscript. We reviewed the manuscript to address the reviewer's comments which improved the quality of the submitted manuscript. We provide further details to each of the individual comments below.
Comment #1: "It is recommended that the authors use the equivalent models shown in Figure 4 and Figure 8 to explain why the proposed WCT is superior to the traditional FBCT in electrical characteristics."
We agree that the originally submitted version of the manuscript had a limited discussion about the sources of the FBCT's imperfections. Although this is not directly evident from the circuit diagrams shown if Figs. 4 and 8, we have expanded on this subject in different sections of the manuscript. We have added a description of previous laboratory studies conducted by one of the authors on FBCT's sensitivity to the beam position and bunch length (lines #154-176 in the reviewed version). We have laid out the decision-making process used for the WCT's conceptual design (lines #196-206 in the reviewed version). We have clarified that the FBCT's design makes analysing its high-frequency behaviour unfeasible (lines #280-285 in the reviewed version). We have further explained the results observed during bunch length sensitivity studies (lines #474-478 in the reviewed version). Finally, we have also expanded the conclusion section and added Tab. 2 to list the most relevant differences between the two sensors.
Comment #2: "The author should sort out all the comparison items and corresponding quantitative data between the proposed WCT and traditional FBCT, and make a table to understand the differences in characteristics."
We agree that a summary table clearly comparing the main characteristics of the WCT and the FBCT was missing in the original version of the manuscript. We have added Tab. 2 in the conclusions which lists the most important differences between the two sensors.
Comment #3: "It is recommended that the author draw the test platform connecting WCT, FBCT, and related test instruments (including equipment models) in section 5 and list all test conditions."
We have added a new figure (Fig. 16 in the revised manuscript) which illustrates how the WCT and the FBCT were characterized using a coaxial line and a Vector Network Analyzer. Moreover the used laboratory instruments are now defined in the lines 396-398 (for time domain measurements) and 417-419 (for frequency domain measurements).
Reviewer 5 Report
Comments and Suggestions for Authors
Measuring beam intensity in a Hadron Collider is a complex scientific and technical problem. Therefore, improving the performance of measuring instruments is an urgent task.
The authors of the paper presented the results of an interesting work on the design and development of a new generation of measurement instruments. The developed WCT successfully eliminated the main performance limitations of the original clot intensity monitor.
The article is well structured and the illustrations are of good quality.
I recommend the article for publication in Energies.
The first reading of the peer-reviewed article made a very good impression. The topic of the article is actual. The authors analyse typical solutions that can be applied for current measurement in such unique installations as Large Hadron Collider. The article shows the disadvantages of these solutions and suggests new more devices that are efficient.
However, a more careful study of the information sources revealed that the article contains a great deal of previously published information. In particular, the article repeats illustrations and text fragments from the reference [12].
Suffice it to cite the following examples. Figures 2, 3, 6, 7, 8, and 9 in the peer-reviewed paper are copies of Figures 2.12, 2.2, 2.3, 3.1, 3.2, and 3.7 in Ph.D. Thesis [12], respectively. Figure 10 repeats Figures 3.8 and 3.9. Figures 11, 12, 13, 13, 14, 15, 16, 17 and 18 in the paper are borrowed from [12]. These are Figures 4.21, 4.24, 4.28, 4.30, 4.12, 4.15, 4.31 and 4.33, respectively.
Therefore, a contradictory situation arises. On the one hand, the main results have already been published. However, they are published in a Ph.D. Thesis, which may be of interest only to a limited circle of specialists.
On the other hand, publication of an article in a well-known journal will find a wide range of readers, which will help popularize the results of interesting work.
Author Response
We thank the reviewer for taking the time to review our manuscript.
We confirm that many figures and results shown in the manuscript were previously published in a PhD thesis defended in 2022. We fully agree with the reviewer that PhD theses, due to their very detailed nature and considerable length, are rather niche documents which are often missed by the broader community. Our goal with writing this manuscript was to disseminate the developed technology within a broader audience who are not exposed to the very specialized and small field of particle beam instrumentation.
We have compiled the most relevant details of the WCT design and presented them in a new way, which we believe might be attractive to the broader readership of Energies. The presented results were chosen to convincingly demonstrate that the developed solution works and to show its potential for applications in domains other that particle beam instrumentation.
To make it more transparent to the editors and the reader that parts of the manuscript describe work which has been published before in a PhD thesis, we added an explicit reference to the said thesis already in the introduction (lines 74 through 80) and included a citation for all figures which appeared previously in the thesis.
Round 2
Reviewer 4 Report
Comments and Suggestions for Authors
Dear Editor,
This manuscript has been revised based on the reviewer's review comments. The content of the manuscript is professional and conforms to the theme of this special issue. It is recommended that it be accepted and published.